# Towards Open-Search De Novo Peptide Sequencing via Mass-Based Zero-Shot Learning

## Abstract

Proteins are the main drivers of biochemical processes and play a pivotal role in almost all cellular functions. Through post-translational modifications (PTMs), residues within a protein can be chemically modified to fine-tune the protein's function in the cellular context. Despite the importance of PTMs, the plethora of deep learning-based de novo peptide sequencing (DNPS) models, which, in contrast to database searching approaches, predict peptide sequence solely from tandem mass spectra without any reference organism database, can only predict peptide sequences with a limited set of PTMs. This is because they rely on fixed vocabularies that map residue tokens to non-generalizable learned embeddings. To overcome this limitation, we propose a novel approach that leverages the fact that amino acids and their derivatives are characterized by their mass, a generalizable feature that enables zero-shot learning. Specifically, we reformulate DNPS as a mass prediction problem instead of a multiclass classification problem, where the model predicts the mass of the next residue instead of its token representation. To facilitate generalization to unseen PTMs, we leverage an adversarial multi-task learning scheme by supplementing the training data of experimental spectra with simulated spectra that mimic spectra containing unseen residues. We show that our approach allows the prediction of previously unseen PTMs, providing a promising proof of concept for mass-based representations as a path towards true open-search DNPS.

## 1 Introduction

Proteins are essential to nearly all biological processes, functioning as catalysts, structural components, signaling molecules, transporters, and immune effectors [1, 2]. Their functional diversity is further enhanced by post-translational modifications (PTMs)—chemical changes to amino acid side chains that influence protein structure and activity and are often linked to disease [3]. Proteomics, the study of proteins, depends on accurate protein identification for downstream analyses such as quantification and interaction mapping [4]. Bottom-up proteomics via liquid chromatography tandem mass spectrometry (LC-MS/MS) is the most common method for high-throughput protein identification [4, 5]. Here, proteins are enzymatically digested into peptides, which are analyzed in a first mass spectrometry scan to determine their mass-to-charge (m/z) ratios. Selected precursor ions are then fragmented, and the resulting fragment ions are measured in a second scan called a tandem mass spectrum [4]. Peptide sequences can, in principle, be inferred from m/z differences between consecutive peaks [6, 7], but this remains challenging due to missing or noisy peaks, co-isolated contaminants, and uncertainty about ion series assignment.

Peptide sequences are typically identified using database search methods, which compare experimental spectra to theoretical spectra derived from a reference protein database [8]. Including PTMs further

expands the space of candidate peptides, making exhaustive searches computationally challenging. De novo peptide sequencing (DNPS) bypasses the need for a reference database by inferring peptide sequences directly from tandem mass spectra. This makes it, in contrast to database search methods, well-suited for identifying novel peptides, rare or unknown PTMs, and proteins from unsequenced organisms. Several de novo peptide sequencing (DNPS) tools have been introduced in recent years, demonstrating promising performance across benchmark datasets [9–25]. However, accurately sequencing post-translationally modified (PTM) peptides remains a significant challenge. Despite the existence of over 400 known PTM types [26, 27], Casanovo—the first transformer-based de novo peptide sequencing (DNPS) model—can identify only seven [11]. Improving PTM prediction has attracted considerable research interest, with recent advances such as AdaNovo [9]. Another approach is to expand model coverage by enlarging the PTM vocabulary; for example, $\pi$-PrimeNovo is fine-tuned to recognize 21 PTMs [28]. However, to the best of our knowledge, no current DNPS model can detect PTMs not seen during training in a zero-shot manner, restricting their ability to discover novel or rare PTMs.

In this work, we propose a transformer-based mass prediction approach to peptide sequencing, enabling zero-shot inference for unseen residues, including novel PTMs. To improve generalization, we explore multi-task learning (MTL) with a training strategy combining experimental and simulated spectra. In the experimental spectra, the model encounters high spectral complexity with a limited set of PTMs, while in the simulated spectra, it learns to generalize to unseen PTMs with arbitrary peak differences. We introduce a generative adversarial network (GAN)-inspired MTL model, which demonstrates strong performance on simulated data and limited but encouraging generalization to unseen residues on experimental spectra. While not solving zero-shot residue inference on experimental data, we provide a framework for future research and highlight key challenges.

## 2   Background and Related Work

In bottom-up proteomics, proteins are extracted from a biological sample and enzymatically digested—typically with trypsin—into smaller peptides [4]. These peptides are separated by liquid chromatography (LC) and introduced into a tandem mass spectrometer (LC-MS/MS), typically operated in data-dependent acquisition (DDA) mode. In the first MS stage (MS1), peptide ions are detected to determine their precursor masses and intensities. The most intense precursors are selected for fragmentation in the second MS stage (MS2), generating spectra composed of fragment ion peaks (tandem mass spectra).[5] The mass differences between these peaks correspond to the masses of individual (modified) amino acids, enabling peptide sequence inference. [6, 7] Peptides are then identified either by matching spectra to theoretical peptides from a reference proteome with database search methods [29] or by interpreting spectra directly with de novo peptide sequencing (DNPS) methods. Identified peptides are subsequently used to infer protein presence and abundance in the original sample. [30]

In the early days of DNPS-based proteomic studies, spectra were manually annotated by experts—a process that was both time-consuming and expensive [30]. As LC-MS/MS throughput increased, early computational tools such as PEAKS [31], NovoHMM [32], and PepNovo [33] were developed to automate de novo peptide sequencing. However, despite their theoretical potential, these methods were often limited in accuracy and struggled with noisy or complex spectra, making database search methods the preferred choice for many applications.

The field has gained renewed momentum with the rise of deep learning. DeepNovo [34] first combined CNNs and LSTMs to model the peptide sequencing process directly. Following this, PointNovo [25] employed a pointer network approach to enhance the decoding of peptide sequences. Transformer-based architectures then marked a significant leap in performance, with Casanovo [11] introducing a streamlined, attention-based model that boosted both accuracy and inference speed. Several other transformer-based tools have emerged since [9, 10, 12, 17, 18, 24, 35], notably AdaNovo, which incorporated adaptive learning strategies to improve performance on peptides containing post-translational modifications (PTMs). Additionally, GraphNovo [16] introduced a graph-based approach to capture the relationships between peptide fragments, further improving sequencing accuracy and handling complex modification patterns. ContraNovo [12] enhanced the transformer decoder of Casanovo by incorporating mass information directly into the model, allowing it to better utilize the mass differences between peptide fragments and improve the accuracy of

peptide sequence prediction, particularly for spectra with ambiguous or overlapping peaks. Recently, $\pi$-PrimeNovo was proposed, a model fine-tuned on 21 PTMs, to expand PTM coverage [24].

These advancements have greatly enhanced the accuracy, generalization, and speed of DNPS tools, making them more viable for applications with incomplete reference databases. However, DNPS models capable of generalizing beyond the amino acids and PTMs present in the training data are still lacking, limiting their ability to accurately discover novel peptides or uncharacterized PTMs.

# 3   Methods

## 3.1   Datasets

### 3.1.1   Experimental Datasets

We use the MassIVE Knowledge Base spectral library v1 (MassIVE-KB v1), originally introduced by Casanovo for DNPS, to train and evaluate our method [11, 36]. This extensive dataset consists of high-resolution HCD mass spectrometry data from diverse experimental conditions, totaling 30,504,897 peptide-spectrum matches (PSMs) with extremely stringent false discovery rate (FDR) control [11]. To facilitate direct comparison with Casanovo [11], we employ the same train, validation, and test splits used in their study. The dataset includes peptide sequences composed of the 20 canonical amino acids, along with eight post-translationally modified amino acids.

### 3.1.2   Simulated Datasets

To establish a model with zero-short learning capacity, we simulated synthetic spectra with a wide variety of possible masses. We reasoned that the simulated masses did not have to match the masses of existing PTM-amino acids. This incentivized models to capture the mechanisms of mass spectrometry rather than memorizing predefined sets of masses.

We first generated peptide sequences as lists of residue masses, each sampled uniformly from the range [60, 300] Da. Peptide lengths were drawn from a uniform distribution between 5 and 20 residues, and precursor charges $z$ were sampled from a discrete distribution: $z = 1$: (0.5), $z = 2$: (0.25), $z = 3$: (0.125), and $z = 4$: (0.125).

We implemented two strategies for generating the corresponding tandem mass spectra. In the first, simplified approach, we generated an ideal spectrum consisting of one peak per residue, corresponding directly to its mass over charge ratio (m/z). The second, more realistic strategy simulates peptide fragmentation by generating b- and y-ions. More specifically, let $z$ be the precursor charge and $\mathbf{M}_{\text{Pep}} = (m_1, m_2, \ldots, m_l)$ the vector of masses for a given peptide of length $l$, where $m_i$ is the mass of the $i$-th residue. The m/z ratios for b-ions are defined as $b_i = \frac{1}{z} \sum_{j=1}^{i} m_j + \text{H}$. Analogously, the m/z values for y-ions are defined as $y_i = \frac{1}{z} \sum_{j=l-i+1}^{l} m_j + \text{H}_2\text{O} + \text{H}$. H and $\text{H}_2\text{O}$ represent the mass of Hydrogen and Water molecules, respectively. To introduce variability, a random number (uniformly drawn from 0 to 5) of peaks were removed from each ion series, while ensuring that at least one peak (b- or y-ion) per residue was retained to preserve full sequence information.

To simulate more realistic spectra, we added between 0 and 10 noise peaks per spectrum, with m/z values sampled uniformly from the range $[50, m/z_{max} + 300]$, where $m/z_{max}$ is the maximal m/z in the simulated spectrum. Fragment ion intensities were sampled from a Gaussian distribution $\mathcal{N}(1.0, 0.1)$, while noise peak intensities were drawn from $\mathcal{N}(0.4, 0.1)$.

## 3.2   Model Architecture

### 3.2.1   Transformer-Based Mass Prediction

Our model builds upon the transformer architecture introduced in Casanovo [11], but reformulates the peptide sequencing task from next-token classification over (modified) amino acid tokens to a continuous mass prediction task. For this, we introduce the mass regression decoder, which outputs a scalar value representing the mass of each peptide residue in Dalton. We implement this by extending Casanovo's peptide decoder with a four-layer feed-forward network with PReLU activation and a single output dimension.

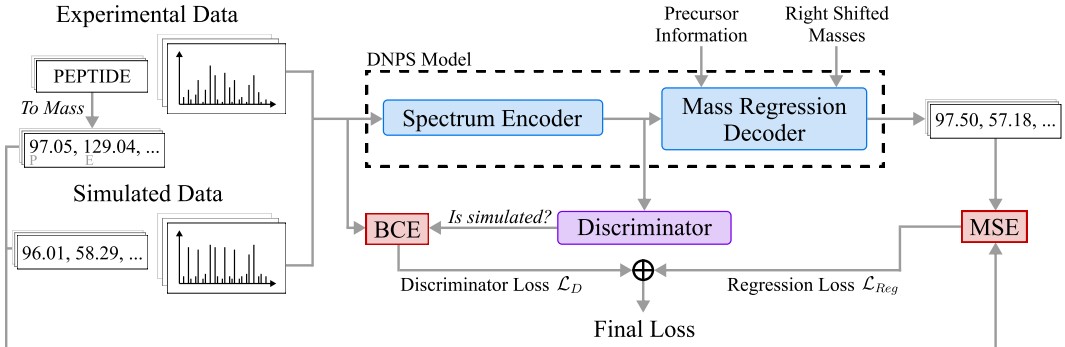

Figure 1: **Adversarial multi-task Learning model architecture for mass-based peptide identification.** Our model uses an encoder-decoder transformer to predict peptide sequences from mass spectra. The decoder predicts the next residue mass based on the spectrum encoding, precursor mass, and previous predictions. During training, the model alternates between experimental and simulated spectra, and is optimized using MSE loss. An adversarial discriminator, trained with BCE loss, encourages domain-invariant encodings by distinguishing simulated from experimental spectra.

For each prediction step, the decoder is conditioned not only on the encoded spectrum and precursor information (mass and charge) but also on the masses of previously predicted residues and the remaining precursor mass, similar to [12]. This setup enables autoregressive decoding of arbitrary residue masses consistent with the spectrum.

To map predicted masses back to residue tokens, we use an extendable mass lookup table, similar to the approach in Contranovo [12]. Specifically, we map each predicted mass to the PTM–amino acid combination that most closely matches its value. Token probabilities are computed via a softmax over the negative absolute differences between the predicted mass and each entry in the table. By extending this set post-training (e.g., by incorporating additional PTMs), the model can predict tokens beyond those encountered during training.

### 3.2.2 GAN-Inspired Latent Alignment

To align representations of experimental and simulated spectra in a shared latent space, we adopt a generative adversarial network (GAN)-inspired framework. The generator in this setup is the transformer encoder, which encodes input spectra into latent vectors. These representations are mean-pooled and passed to a discriminator—a three-layer feed-forward neural network (FNN) with Leaky ReLU activations—adapted from the discriminator used by Wu et al. [37]. The discriminator is trained to distinguish between embeddings of experimental and simulated spectra, while the DNPS model is adversarially regularized by the discriminator's loss, thereby encouraging the encoder to learn modality-invariant representations.

### 3.3 Mulit-Task Training Strategy

We train the model on a mixture of simulated and experimental spectra, sampling balanced batches from both sources. The mass prediction model is optimized using a mean squared error (MSE) loss $\mathcal{L}_{Reg}$ between predicted and ground truth residue masses. To train the adversarial extension of the multi-task learning (MTL) model, we incorporate an additional regularization term based on the discriminator's binary cross-entropy (BCE) loss $\mathcal{L}_D$. The MTL model is trained on a composite loss $\mathcal{L}_{Adv}$, defined as a linear combination of the mass regression loss $\mathcal{L}_{Reg}$ (weight 1) and the discriminator loss $\mathcal{L}_D$ (weight -50).

The MTL model weights are initialized with weights from a model pre-trained solely on simulated spectra from our simplified simulation approach for 80,000 steps. We train for 600,000 steps with a batch size of 64 (approximately 1.5 epochs) on a single A40 GPU with 8 CPU cores and 60 GB of CPU RAM for approximately 2 days. The learning rate is set to $4 \times 10^{-4}$ for the mass prediction DNPS model and $1 \times 10^{-6}$ for the discriminator. The validation set was evaluated every 50,000 training steps, and the final model corresponds to the checkpoint with the lowest validation loss.

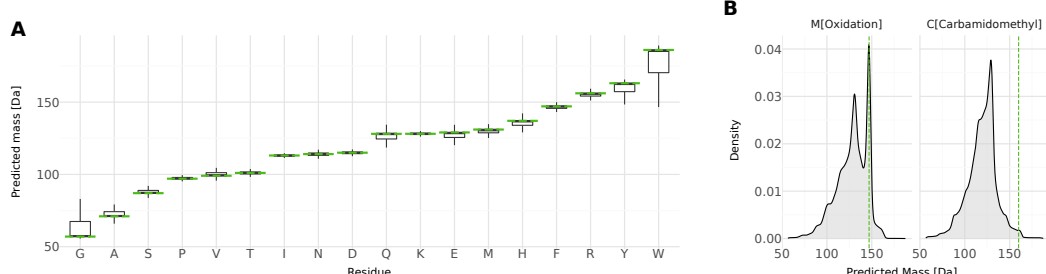

Figure 2: **Mass regression transformer evaluation on seen and unseen PTM-amino acid combinations during training. A**, Predicted masses on test set for all amino acids present in the training split of the MassIVE-KB V1 dataset. The horizontal dashed lines indicate the target mass, i.e., the true mass of the residue. Outliers are not displayed. **B**, Distribution of the predicted mass for the two PTM-amino acid combinations, methionine oxidation and cysteine carbamidomethylation, withheld from the training split of the MassIVE-KB V1 dataset. The vertical dashed lines indicate the target masses.

## 4 Experiments

To enable de novo peptide sequencing (DNPS) models to generalize beyond the set of amino acids and post-translational modifications (PTMs) observed during training, we reformulated the task as a continuous mass prediction problem. Instead of selecting residues from a fixed vocabulary, our model predicts a scalar mass for each peptide position, allowing for the potential inference of arbitrary or novel modifications. These predicted masses are then mapped to residue identities via a flexible, extendable lookup table. To encourage generalization, we adopt a multi-task training strategy that combines complex experimental spectra, limited to a known set of PTMs, with simulated spectra engineered to include a broad range of synthetic modifications. Additionally, we introduce a GAN-inspired architecture, which aims to align latent representations of real and simulated spectra and improve the model's ability to bridge between the two domains.

### 4.1 Evaluation Metrics

To assess the model's ability to generalize to modified amino acids not seen during training, we excluded all spectra containing methionine oxidation and cysteine carbamidomethylation from the training and validation sets—effectively treating these modifications as unseen during evaluation.

We evaluated performance at two levels: mass accuracy and residue identification. Mass accuracy was measured as the absolute difference between the predicted and ground truth masses for each residue. Residue identification was assessed by mapping each predicted mass to the nearest entry in a lookup table containing all amino acid and PTM combinations present in the MassIVE-KB v1 dataset. The predicted residue was then compared to the ground truth to compute recall at the amino acid level.

For evaluation, we ran the model in inference mode using teacher forcing. In this setup, the model is supplied with the ground truth masses of all previously decoded residues at each prediction step. This prevents error accumulation from incorrect predictions and isolates the model's ability to predict each residue mass independently. Additionally, by enforcing the correct number of decoding steps, teacher forcing ensures that the predicted and ground truth residue sequences are aligned, facilitating direct comparison of their respective masses.

### 4.2 Main Results

#### 4.2.1 Mass Regression Decoder Confidently Predicts Masses Seen During Training

Our results indicated that recasting peptide sequencing as a mass prediction task preserved the model's ability to infer residue identities. The model reliably predicted the masses of unmodified residues seen during training on the defined test set (Fig. 2A). Across all unmodified residues seen during training, our model achieved a median absolute error of approximately 0.56 Da on the test

set (Pearson correlation coefficient: 0.89, P-value<0.05). Errors ranged from 1.10 Da for tryptophan (W) to 0.39 Da for glycine (G). This precision allowed the model to distinguish residues effectively, resulting in an overall amino acid-level recall of 62.37%. Because we assigned predicted masses to the closest matching residue in a lookup table, the model performed better for amino acids that were well-separated in mass. For example, arginine (R) was correctly recalled in approximately 79.91% of cases. In contrast, the model struggled with residues whose masses are close to others, such as lysine (K), which had a recall of only 26.97%—despite having a lower median error (0.48 Da) than arginine (0.61 Da). While these recall values were somewhat lower than those reported by classification-based transformer models such as AdaNovo [9], this was expected given the nature of our approach. Predicting scalar masses imposes a stricter requirement for precision: small deviations could lead to mismatches when mapping back to discrete residue tokens. In contrast, models like Casanovo [11] benefited from the flexibility of learned embedding spaces, where similar residues could be placed further apart to ease classification. Despite this inherent challenge, our results demonstrated that the mass-regression decoder effectively learned accurate mass representations for residues seen during training, providing a viable foundation for generalization to modified residues beyond the training set.

While the model successfully predicted the masses of residues seen during training, capturing a generalizable and interpretable biochemical feature, it struggled to generalize to truly novel modifications (Fig. 2B). For example, predictions for cysteine carbamidomethylation showed no enrichment near the correct mass, with the distribution shifted toward lower values and a recall of only 0.6%. A contributing factor to the failed extrapolation might be that all cysteines in the data were modified, meaning the model never encountered an unmodified cysteine during training. In contrast, predictions for methionine oxidation were modestly enriched around its correct mass, achieving a recall of 14.18%. However, this perhaps reflected a memorization artifact: the mass of methionine oxidation ($\sim$147.04 Da) was only $\sim$0.03 Da less than phenylalanine (F), a residue included in training, indicating the model might simply be reproducing familiar masses. The observed bias toward masses seen during training was consistent with trends in generalized zero-shot learning [38]. Since the training objective does not explicitly encourage extrapolation to unseen masses, the model was not incentivized to learn a truly continuous mass space. Instead, it might implicitly treat the task as a form of multi-class classification, where the "classes" were the residue masses encountered during training. As a result, the model could memorize these masses and reproduce them at inference time, without being penalized by the loss function for failing to predict novel ones.

### 4.2.2    Mass Regression Model Can Solve Simulated Open-DNPS Problem

To mitigate the model's bias toward training-set residue masses—an issue arising from the limited diversity of modifications in experimental spectra—we explored the use of simulated spectra spanning a broader amino acid–PTM search space. In these simulations, peptides were constructed by sampling random residue masses to mimic modified amino acids, thereby preventing memorization. We designed three simulation levels of increasing complexity and trained a separate model on each. Across all settings, the model successfully learned to predict the correct masses (Fig. 3A, B), though performance declined as simulation complexity increased (Pearson correlation coefficients of 1.0, 0.99 and 0.97 with two-sided P-values<0.05). Introducing a variable number of peaks led to higher mass prediction errors compared to simulations with a fixed number of peaks (Fig. 3B), as expected from the increased difficulty. Interestingly, the model remained robust when noise peaks and realistic fragmentation processes were introduced. However, these settings caused the MSE loss to more than double, largely due to occasional large prediction errors (108.96 vs. 263.98). This is likely because the simulation emulated missing peaks (although never both of the complementary b- and y-ion), causing the model to infer residue masses from unrelated fragments, leading to strong deviations and quadratic penalties from the MSE loss.

Although the model performed well on simulated spectra, it failed to generalize to experimental spectra and accurately predict real residue masses (Pearson correlation coefficients of -0.11, 0.19 and 0.04 with two-sided P-values<0.05). We reasoned that the domain shift between simulated and real data was too substantial, leading models trained on simplified simulations to break down entirely when applied to real spectra (Fig. 3C). These models produced nearly identical mass distributions across all residues, indicating a lack of residue-specific signal. The model trained on the most realistic simulation—incorporating both noise peaks and fragmentation mechanics—showed some limited signs of generalization to experimental spectra (Fig. 3C). It produced slightly differentiated mass

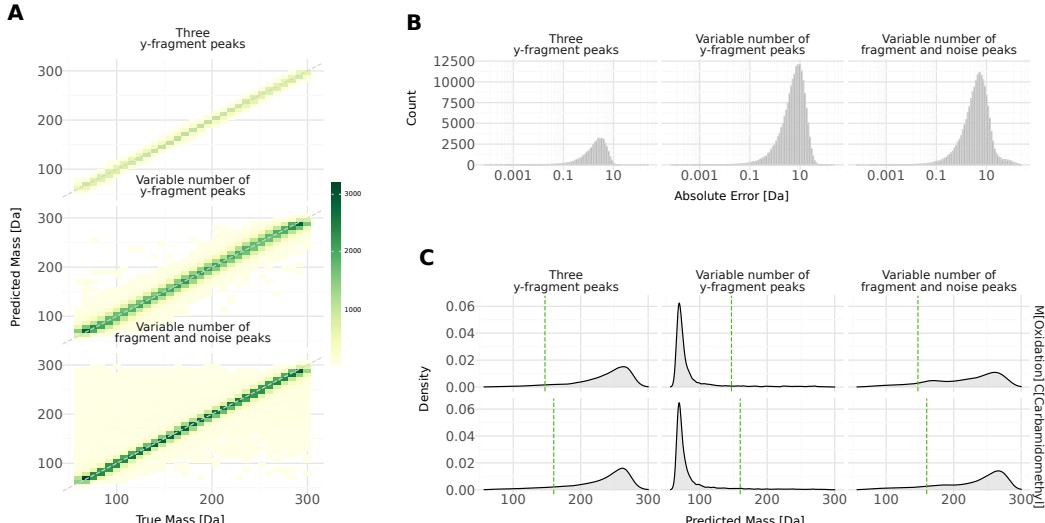

Figure 3: **Mass regression transformer on simulated data**. **A**, Predicted against true underlying simulated residue masses for three data simulation strategies (facets): (i) spectra with three peaks with arbitrary mass distance and precursor charge=1. (ii) a variable number of peaks with arbitrary mass distances and precursor charge=1 (iii) spectra with a variable number of peaks corresponding to b- and y-fragments, precursor charges>1, and noise peaks. **B**, Absolute errors between the predicted masses and true masses for the different data simulation strategies (facets). **C**, Distribution of predicted masses for experimental data from MassIVE-KB V1 containing methionine oxidation and cysteine carbamidomethylation residues. The vertical dashed lines indicate the target masses.

260 distributions for individual residues, with subtle enrichments around masses offset by approximately
261 18 Da from the target values. This offset aligned with the mass of a water molecule (∼18 Da),
262 which distinguishes y-ions from b-ions, and may reflect confusion between ion series in experimental
263 spectra. While this suggested that the model was extracting some transferable features from the
264 realistic simulations, the extent of generalization remained minimal. Performance on experimental
265 data remained inadequate, with recall for cysteine carbamidomethylation and methionine oxida-
266 tion residues reaching only 0.84% and 1.19%, respectively. These results highlight that, although
267 realistic simulation improved alignment with experimental characteristics, simulated data alone
268 were insufficient for teaching the model to handle the complexity of real-world spectra and unseen
269 modifications.

### 4.2.3 Multi-Task Learning Improves Generalization

271 We explored Multi-Task Learning (MTL) as a strategy to improve the model's ability to generalize,
272 particularly to unseen post-translational modifications. MTL has proven effective in various domain
273 adaptation contexts [39–41], and we adapted it by training the model on a balanced combination of real
274 and simulated spectra. The rationale was twofold: training on real experimental spectra encourages the
275 model to learn the inherent complexity and noise characteristics of true mass spectrometry data, while
276 training on simulated spectra—which include a diverse set of unrestricted residue masses—pushes
277 the model to generalize beyond the limited set of modifications seen in the real data. By combining
278 both sources, the model was encouraged to learn features that were robust across domains while
279 being flexible enough to infer arbitrary modifications.

280 The MTL model improved performance on simulated spectra (Pearson correlation coefficient: 0.98
281 with two-sided P-value<0.05, Fig. 4A) but sacrificed precision on seen residues in experimental spec-
282 tra compared to the model trained exclusively on experimental data (Pearson correlation coefficient:
283 0.83 with two-sided P-value<0.05, Fig. 4B, median absolute errors of 3.17 Da vs. 0.56 Da). On
284 the other hand, the MTL model outperformed the model trained solely on the simulated data with
285 a lower median absolute error of 2.95 Da compared to 4.25 Da. The imbalance between the MTL
286 model's performance on simulated and experimental spectra might stem from the loss formulation:

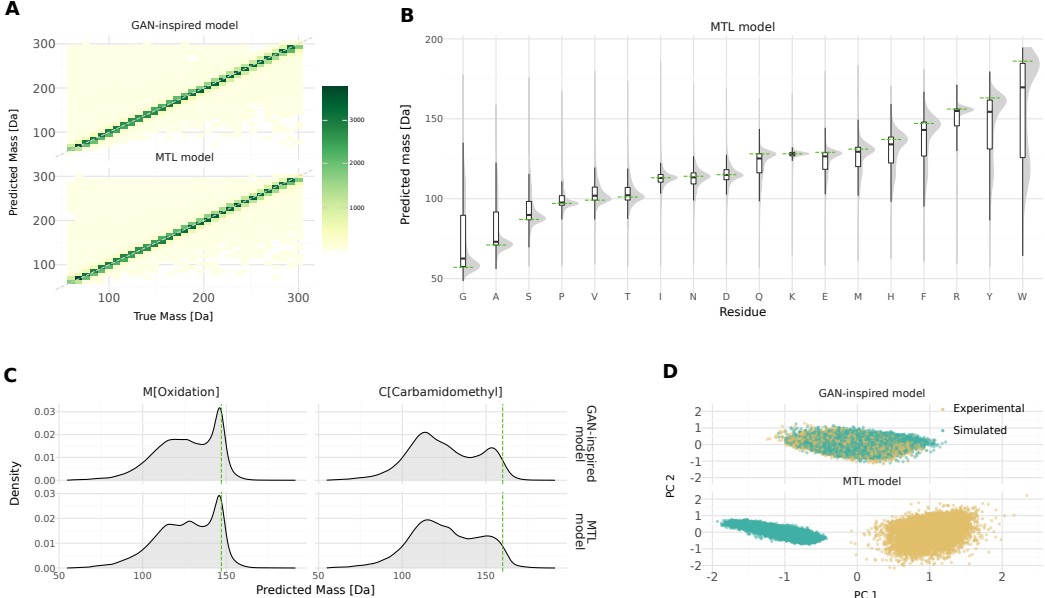

Figure 4: **Multi-task learning and adversarial-inspired model for PTM-amino acid mass prediction**. **A**, Predicted against the true masses on simulation data for the two different models (facets): (i) model using multi-task learning scheme with a mixture of experimental and simulated spectra and (ii) extended with an adversarial loss term that is obtained by training a discrimination module that predicts whether a spectrum is simulated or not. **B**, Distribution of predicted masses for all amino acids seen during training. The horizontal lines indicate the true target masses. **C**, Distribution of the predicted masses of the two unseen residues, cysteine carbamidomethylation and methionine oxidation, which were withheld from the training data. The vertical lines indicate the target masses. **D**, Two-dimensional PCA projection of pooled spectrum encoder embeddings for the two different models (facets).

since simulated spectra were easier to learn from, the model might prioritize reducing their loss, which offered a more efficient path to minimizing the overall objective, potentially at the expense of generalizing to real spectra. This behavior contrasted with the ideal MTL outcome, where both tasks mutually benefit from shared representation learning [42].

While the MTL model showed improved performance on simulated spectra, this came at the cost of reduced accuracy on known residues in experimental data. However, this trade-off coincided with emerging signs of generalization to unseen residues excluded from training (Fig. 4C). Notably, the peak near the target mass for methionine oxidation was more distinct and sharper than in the model trained without MTL (Fig. 2B and Fig. 4C), indicating increased confidence in these predictions. For the unseen cystein carbamidomethylation, the MTL model's predicted masses shifted closer to the target mass, with a pronounced peak just below it—contrasting sharply with the model trained only on experimental data. This suggests that MTL enables the model to better generalize and recover evidence for unseen modifications that were previously overlooked.

Despite improved generalization, the MTL model still showed some bias toward masses seen during training, though to a lesser extent. For instance, the secondary peak in predicted masses for cystein carbamidomethylation aligned with the amino acids (iso)leucine (I, L ~113 Da), asparagine (N ~114 Da), and aspartic acid (D ~115 Da). Consequently, absolute prediction quality for unseen residues remained limited. While recall for cystein carbamidomethylation improved fivefold compared to the model trained only on experimental data, it remained low at about 3.18%. Recall for methionine oxidation increased only marginally, from 14.18% to 14.41%.

### 4.2.4 Adversarial Loss For Common Latent-Space For Spectrum Embedding

Although the MTL model performed well on simulated data, it was still biased toward seen masses. To investigate whether simulated data was effectively leveraged during training, we analyzed the embeddings with a 2D PCA projection (Fig. 4D). The embeddings for experimental and simulated spectra formed distinct clusters, suggesting that the transformer-based mass regression decoder could easily differentiate between the two data types. This separation might have caused the model to treat simulated spectra differently, reducing their effectiveness in helping the model generalize to realistic, unseen masses.

To reduce the separation between simulated and experimental spectra, we incorporated adversarial learning by adding a lightweight binary classifier to our MTL model that predicted whether a spectrum was simulated or experimental based on its encoding. The classification loss was subtracted from the MTL model's MSE loss to encourage a shared latent space for both types of spectra. Following the addition of this adversarial loss, the embeddings of simulated and experimental spectra no longer separated clearly in PCA space (Fig. 4D). Moreover, the discriminator's binary cross entropy loss during training plateaued at around 0.68, meaning its predicted probabilities lay consistently around 0.5 ($-\log(0.5) \approx 0.69$). However, the loss and PCA alone did not provide definitive evidence that the model could not still distinguish between the two spectrum types. This was further supported by the discriminator achieving an area under the ROC curve of around 0.78.

Unfortunately, the addition of the adversarial loss term did not lead to a noticeable improvement in the MTL model's ability to generalize to unseen residues. The distributions of predicted masses for unseen residues remained very similar between the MTL models with and without adversarial loss (Pearson correlation coefficient: 0.82 with two-sided P-value<0.05, Fig. 4C). Additionally, both models showed comparable recall rates for methionine oxidation (3.18% vs. 3.10%) and cystein carbamidomethylation (14.41% vs. 15.11%), along with similar MSE losses.

## 5 Conclusion and Future Work

We present a novel framework for de novo peptide sequencing that reformulates the task as mass regression rather than discrete classification. This shift enables the model to move beyond a fixed vocabulary of amino acids and modifications, allowing it to predict residues not encountered during training. Our approach combines transformer-based mass prediction with a multi-task learning setup trained on both experimental and simulated spectra. To encourage domain-invariant representations, we incorporate adversarial learning that aligns the spectrum encodings across data sources. Results demonstrate promising generalization to previously unseen modifications, marking a step toward more flexible and open-ended peptide sequencing. By enabling the discovery of novel or rare peptide modifications, this work may support future advancements in biomedical research.

**Limitations.** This work serves as a proof of concept rather than a production-ready system. Although we show that mass-based modeling can overcome vocabulary constraints, the current framework has only been evaluated under teacher-forced decoding. Nonetheless, it provides a foundation for developing fully open de novo sequencing models that support more reliable and practical applications.

**Future directions.** Several avenues merit further investigation: exploring the tradeoff between scalar mass prediction and vector-based encodings, improving robustness to noise in simulated spectra, and developing loss functions that better prioritize precision. The role of adversarial learning also warrants deeper analysis, particularly whether the model implicitly treats simulated data differently. Incorporating high-fidelity simulated spectra (e.g., from models like Prosit [43, 44]) and more advanced decoding strategies could further improve performance. Together, these advances could enable DNPS models that generalize across biological contexts, capturing both known and novel peptide modifications with greater reliability.

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
