# OpenReview forum: "Towards Open-Search De Novo Peptide Sequencing via Mass-Based Zero-Shot Learning"
_NeurIPS.cc/2025/Conference — Submitted to NeurIPS 2025_

### Official Review · Reviewer_SmV5 · 2025-06-07

**Clarity:** 2
**Significance:** 1
**Originality:** 3
**Rating:** 2
**Confidence:** 5

**Summary:**

The prediction of post-translational modifications (PTMs) on peptides is important due to their vital biological functions. Existing DNPS models can only predict a limited set of PTMs that are present in the training data. This paper proposes a novel approach that reformulates PTM prediction as a mass prediction problem, which allows for the identification of previously unseen PTMs.

**Questions:**

As noted in the weaknesses section, my primary questions for the authors are:
1. Why were no performance comparisons made against any existing DNPS models, even on tasks where such comparisons would be directly applicable?
2. Why does the evaluation lack peptide-level metrics, such as peptide precision, given that these are the most critical indicators of a model's practical utility for the DNPS task?

3.The presentation of Section 4.2 seems to follow a progressive, step-by-step structure that introduces the model's components sequentially. Why was this framed as a series of 'main results' rather than as a formal ablation study, which is typically used to demonstrate the individual contribution of each component?

**Ethical Concerns:**

["NO or VERY MINOR ethics concerns only"]

**Final Justification:**

i maintain the rating 2 (Reject) with high confidence (5) because:

- use of teacher forcing during the testing phase, a fact the authors clearly acknowledge (major)
- absence of performance comparisons against any baseline DNPS models (major)
- The model's reported experimental performance is underwhelming (major)
- chose two of the most common and data-rich PTMs
- Lack of Comprehensive Metrics

The progress made in this paper appears is limited. Some preliminary explorations and attempts are not quite up to the acceptance standards of a top-tier avenue

**Limitations:**

I want to state clearly that the problem this paper focus—open-search de novo peptide sequencing (DNPS) for unknown PTMs—is a critically important and exceptionally difficult task in peptide sequencing and proteomics. However, this manuscript reads more like a cursory experimental report than a substantive, innovative contribution with practical significance.

This paper undertakes a preliminary exploration into the important field of open-search DNPS and presents a series of experimental results. Although the proposed model is novel, its performance is poor, and its methodology is critically flawed by the use of teacher forcing during the inference stage.

**Quality:**

1

**Strengths And Weaknesses:**

**strengths：**

1.Nearly all previous DNPS models are incapable of predicting PTMs unseen during training. This is an inherent limitation of their architecture, as they can only identify PTMs included in a predefined 'vocabulary'. In contrast, the model proposed in this study is a rare example for performing open-search DNPS.


2.The approach presented in this paper—reformulating DNPS as a mass prediction problem instead of a multiclass classification problem—is both novel and promising.

**weaknesses：**

1.My primary concern is the model's use of teacher forcing during the testing phase, a fact the authors clearly acknowledge. In this setup, the model predicts the remainder of a peptide based on a partial ground truth sequence. This is a severe form of label leakage, rendering the model inapplicable to any practical DNPS scenario, where ground truth information is entirely unavailable.

It is a fundamental consensus in the deep learning community, i think, that the testing process must be completely free from such direct label leakage.


2.Another major concern is the complete absence of performance comparisons against any baseline DNPS models; the paper only presents its own model's results in isolation. While existing baselines may not support the task of open-search DNPS, they could and should have been included in several experiments for a fair evaluation.

For example, the task of "Mass regression evaluation on seen amino acid combinations" (Section 4.2.1, Figure 2A) is unrelated to the open-search. A direct comparison could have been performed in one of two ways: by converting the discrete amino acid outputs of baseline DNPS models into scalar mass values, or by converting this proposed model's scalar mass output back into discrete labels (which the authors explicitly state is possible). Similarly, baseline models could have readily participated in the experiments presented in Section 4.2.3 (Figure 4B) and those involving simulated data in Section 4.2.2, either by training or testing them on the same data.

Without any performance comparison against existing DNPS models, it is difficult to ascertain the true contribution and advancement presented in this paper.

3.Given that the main innovation is the ability to predict unknown PTMs, the experiments should have logically focused on rare, low-data modifications not covered by previous DNPS models. Instead, the authors chose methionine oxidation and cysteine carbamidomethylation for their experiments—two of the most common and data-rich PTMs that nearly all existing DNPS models already predict.

As a result, the experiments is not solid to demonstrate the model's most crucial selling point: its claimed ability to perform open-search DNPS for unknown modifications.


4.Lack of Comprehensive Metrics. The evaluation metrics include 'mass accuracy' and 'residue identification' but neglect the most critical metric for DNPS tasks: peptide-level precision. This metric serves as the principal indicator of a DNPS model's practical utility, as the ultimate goal of DNPS is to assign complete and accurate peptide sequences to each mass spectrum. After all, a correctly identified unknown PTM holds little value if the overall peptide sequence is incorrect.

5.The model's reported experimental performance is underwhelming. For instance, the authors acknowledge that the "absolute prediction quality for unseen residues remained limited," with the recall for cysteine carbamidomethylation being as low as 3.18% and for methionine oxidation only 14.41%. This underwhelming performance is also visually evident. Figures 2A, 3C, and 4D clearly show that when the model predicts unknown PTMs, the distribution of its predicted masses deviates significantly from the actual ground truth mass of the PTM.

---

> ### Author Rebuttal · Authors · 2025-07-31
>
> Firstly, we want to thank the reviewers for their constructive comments on our paper. Since we attempted to tackle a difficult problem in DNPS, the community’s feedback is critical to us, and we genuinely appreciate it.
>
> ## General Remarks
>
> Multiple reviews have commented on two points: the practical applicability of our approach and the missing comparisons to other methods, such as Casanovo.
>
> Regarding the practical utility of our approach, we want to take this opportunity to restate very clearly that we do not claim to have a full model that can be practically applied yet. The problem of open search DNPS is truly hard, as is acknowledged by one reviewer. We believe to have found an approach that could potentially lead the DNPS community (over time) towards true open search models. We appreciate that one reviewer has recognized our novelty. We see our paper as a first step and mentioned this transparently in the paper’s limitations section. As a result, we did not provide peptide-level performance metrics, as our contribution is not the improved prediction of full peptides, but rather to demonstrate a first step towards the generalization to unseen residues. However, we do see that this point can be emphasized more clearly throughout the paper, and we will add further clarification to the camera-ready version once possible.
>
> Secondly, we want to briefly discuss why we do not compare our model's performance against that of other methods, e.g., Casanovo. Our paper’s contribution is that of generalizing to residues not present during training. Thus, we would primarily be interested in comparing these predictions against Casanovo’s performances. However, if we were to construct a fair setting and also remove these residues from Casanovo’s training (and remove the vocabulary), its recall would, by construction, be 0. This is because Casanovo and all other comparable tools cannot predict residues not included in their training sets. Thus, these comparisons do not yield any valuable insight and were purposefully omitted in our paper.
>
> In light of the above general clarification, we respectfully ask the reviewers to revisit their assessment of our paper in the context of zero-shot generalization.
>
> ## Reviewer Concerns
>
> ### Concern 1
>
> The reviewer’s first concern touches upon the use of teacher forcing for evaluating our model’s capability to predict unseen residues. We want to point out that we only use teacher forcing for the model’s evaluation, but do, of course, not intend to use it in prospective applications. However, as mentioned above, our paper tries to provide the DNPS community with a potential way towards open sequencing, and in the current state, we only want to show the general possibility to predict residues that were not seen during training. We have found this to be most consistently done by using teacher forcing, which decouples the impact of previous prediction errors from the model's performance to predict (unseen) residues. Also, we want to stress that in the way we evaluate the model (which is AA-level metrics exclusively), teacher forcing does not constitute label leakage. Since we for each prediction only provide the previous predictions, the model cannot leak the current AA identity and thus will not affect the current AA’s prediction.
>
> Nonetheless, future work continuing to explore the direction of open DNPS will have to move to peptide-level metrics - once practical applicability becomes of concern.
>
> ### Concern 2
>
> The reviewer’s second concern regarding model comparison is very valid, and we hope to have addressed parts of it above. The reviewer is right, however, and comparisons to other models in terms of, e.g., AA recall for residues seen during training would add insight, and we should have added those. Because of the short rebuttal, we currently do not have these values in teacher forcing mode for other models, but our own.
>
> Nonetheless, we want to stress that these comparisons would then only aim to show that our model does not totally fail in these cases and manages to get a reasonable AA recall (as done in lines 205ff.). We particularly would not aim to outperform any method, since our mass prediction problem in one dimension is much harder (compared to Casanovo’s residue identity prediction) and will lead to inevitable performance costs.
>
> Finally, we want to point out that it would not be easily possible to train classical DNPS models on simulated data. Our simulated spectra are specifically designed not to be restricted to a fixed set of residue masses. Thus, a fixed vocabulary model, such as Casanovo, is not applicable. The reviewer’s question does, however, raise an interesting question beyond the scope of our paper: To what number of residues can Casanovo’s (or any other model’s) vocabulary be expanded before degrading performance, i.e., could a fixed vocabulary model under perfect simulated data conditions even be applied to hundreds of PTMs?
>
> ### Concern 3
>
> The reviewer’s third concern regards the selection of two common PTMs as our unseen residues and why we did not decide to use rare PTMs with less data. This is a valid question, and our decision comes down again to our paper aiming to be the very first step towards open DNPS and providing a proof of concept. In this paper, our evaluation of peptides with unseen residues aimed to ensure that the corresponding spectra were comparable to the model’s training data, so as not to confound the evaluation with additional challenges typically associated with modified peptides, such as longer sequences, increased spectral complexity, and altered fragmentation patterns. Therefore, we deliberately decided to use hidden residues from the training data set to provide a proof of concept. Nevertheless, we agree that data containing PTMs (such as phosphorylation) must be considered in future research when practical applicability is aimed for. Further, we chose to only hide 2 compounds, as these were the ones that allowed us to remove all peptides containing them without losing too many PSMs.
>
> ### Concern 4
>
> In the above, we hope to have made clear why we deliberately did not provide peptide-level metrics. Regardless, future works moving towards applicable models will have to include peptide-level metrics.
>
> ### Concern 5
>
> The reviewer’s last concern regards the low recall values of the unseen residues and the visually distorted mass distributions of the unseen residues.
>
> We understand that the recall values in absolute terms do appear to be underwhelming. However, we want to stress how difficult residue prediction via mass as a proxy is. We explicitly mention this problem in lines 213ff of our paper: ‘Predicting scalar masses imposes a stricter requirement for precision: small deviations could lead to mismatches when mapping back to discrete residue tokens. In contrast, models like Casanovo [11] benefited from the flexibility of learned embedding spaces, where similar residues could be placed further apart to ease classification.’ To illustrate this problem, I would like to consider the residues I, N, and D with masses of ~113, ~114, and ~115 Da, respectively. For the model to correctly predict an N, it would have to predict the correct mass within 0.5 Da. Thus, even small mistakes will lead to wrong token predictions and depress the peptide recall and precision.
>
> To address this issue, we also proposed a possible avenue in lines 346ff of the paper: ‘exploring the tradeoff between scalar mass prediction and vector-based encodings.’ For example, one could consider not predicting the scalar mass (thus having the problem of very close AA masses), but rather predicting mass encodings - similar to what is used to encode peaks in the spectrum encoders. These vector encodings could still be mapped to a set of residues, similar to what is done in ContraNovo.
>
> Secondly, the reviewer mentions the visually distorted distributions, which do not follow the ground truth distribution. And again, this might seem underwhelming at first. However, by even showing any increased densities around the ground truth, we have shown that he model does indeed extract a signal from spectra for a residue never seen during training. Here, it is again important to stress that our paper is the first of many hard steps towards open DNPS, and we show that by leveraging simulated spectra, one can indeed construct a DNPS model that can generalize beyond a fixed set of vocabularies. Future work is clearly needed to make it practically applicable, which we do honestly acknowledge.

---

> > ### Comment · Reviewer_SmV5 · 2025-08-05
> > **rebuttal**
> >
> > Thank you for the detailed explanation.
> >
> > - Regarding teacher forcing: Unfortunately, I cannot fully agree with the author's response. When performing the DNPS task, for an autoregressive model (such as Casanovo and the model in this paper), the amino acid sequence information of the prefix is crucial, in addition to the input mass spectrum information. Evaluating the model using partial ground truth amino acid information (teacher forcing) provides a significant advantage to the model (otherwise, I believe the authors would not have used it) and, at the same time, constitutes label information that should not be fed to the model (as the goal of DNPS is to predict the complete peptide sequence relying solely on mass spectra).
> >
> > - As for the remaining questions, i believe the authors have merely explained why they did and what they did, but their reasoning is not convincing and they have failed to adequately address the issues through experiments or other means.
> >
> > Although open search is a difficult and meaningful problem, the progress made in this paper appears, in my opinion, limited. Some preliminary explorations and attempts are not quite up to the acceptance standards of a top-tier avenue (like NeurIPS).
> >
> > i appreciate the author's effort in responding to the comments

---

### Official Review · Reviewer_sSSm · 2025-07-01

**Clarity:** 3
**Significance:** 2
**Originality:** 2
**Rating:** 2
**Confidence:** 2

**Summary:**

This paper presents a novel de novo peptide sequencing approach based on mass regression and multi-task learning, aiming to achieve "zero-shot generalization" to unseen post-translational modifications (PTMs). The model is jointly trained on experimental spectra and large-scale simulated spectra, with the introduction of an adversarial discriminator attempting to align the distributions of experimental and simulated domains in the latent space.
While the work introduces promising methodological innovations, more rigorous comparisons against state-of-the-art methods, deeper analysis of adversarial mechanisms, and significantly expanded validation of zero-shot PTM generalization are essential.

**Questions:**

Required Revisions:
1. Systematic baseline comparison: Include quantitative performance tables comparing against Casanovo, AdaNovo, π-PrimeNovo, ContraNovo and so on.
2. Comprehensive generalization assessment: There are “over 400 known PTM types”, please extend PTM generalization testing beyond M[Oxidation] and C[Carbamidomethyl]. And conduct direct comparisons with leading de novo peptide sequencing methods on multiple unseen PTM types.
3. Clarify data partitioning: Explicitly state the criteria for preventing data leakage (e.g., sequence identity thresholds, spectral similarity-based splits) to ensure rigorous evaluation.
4. Detail Multi-task learning mechanisms:
(1) Specify the training relationship between the discriminator and main model (e.g., alternating or joint updates?).
(2) Provide mathematical formula for the composite adversarial loss term (LAdv).
(3) Justify the selection of the adversarial weight (weight -50) with empirical or theoretical rationale.
(4) Ablation studies on adversarial design:
a. Different discriminator architectures.
b. Varying adversarial loss weights.

**Ethical Concerns:**

["NO or VERY MINOR ethics concerns only"]

**Final Justification:**

Thank you for your rebuttal. Your rebuttal addressed the concerns regarding question 4, but the other content has not been fully resolved. I will maintain the original score.

**Limitations:**

1.	The generalization scope is severely constrained: PTM evaluations are limited to only M[Oxidation] and C[Carbamidomethyl], failing to reflect model capability across hundreds of biologically relevant PTMs.
2.	No comprehensive quantitative benchmarking against state-of-the-art models is provided.
3.	The adversarial design choices lack rigorous ablation studies and mechanistic analysis.

**Quality:**

2

**Strengths And Weaknesses:**

Strengths:

1. The methodology demonstrates innovation by reformulating peptide sequence prediction as a continuous mass regression task, overcoming limitations of fixed token-based approaches. This theoretically enables zero-shot generalization potential.
2. The multi-task learning strategy, integrating both experimental and simulated spectra, enhances the model's overall robustness and partial PTM generalization capability.
3. The authors transparently disclose the methodological limitations of their work, demonstrating commendable scientific rigor.

Weaknesses:
1. Lack of systematic quantitative comparison against established baselines (e.g., Casanovo, AdaNovo, π-PrimeNovo, ContraNovo). Only indirect or partial qualitative mentions of one or two baseline models are provided; no comprehensive comparative tables are presented.
2. Limited generalization to novel PTMs (e.g., C[Carbamidomethyl]). Evaluation is restricted to only two common modification types (M[Oxidation] and C[Carbamidomethyl]), lacking broader coverage. Crucially, no direct performance comparison to other baseline methods on unseen PTMs.
3. Insufficient explanation of the adversarial discriminator mechanism. Key design choices—such as the rationale for the adversarial loss weight selection (e.g., justification for -50), and the impact of discriminator capacity on domain alignment and generalization—lack detailed experimental validation.

---

> ### Author Rebuttal · Authors · 2025-07-31
>
> Firstly, we want to thank the reviewers for their constructive comments on our paper. Since we attempted to tackle a difficult problem in DNPS, the community’s feedback is critical to us, and we genuinely appreciate it.
>
> ## General Remarks
>
> Multiple reviews have commented on two points: the practical applicability of our approach and the missing comparisons to other methods, such as Casanovo.
>
> Regarding the practical utility of our approach, we want to take this opportunity to restate very clearly that we do not claim to have a full model that can be practically applied yet. The problem of open search DNPS is truly hard, as is acknowledged by one reviewer. We believe to have found an approach that could potentially lead the DNPS community (over time) towards true open search models. We appreciate that one reviewer has recognized our novelty. We see our paper as a first step and mentioned this transparently in the paper’s limitations section. As a result, we did not provide peptide-level performance metrics, as our contribution is not the improved prediction of full peptides, but rather to demonstrate a first step towards the generalization to unseen residues. However, we do see that this point can be emphasized more clearly throughout the paper, and we will add further clarification to the camera-ready version once possible.
>
> Secondly, we want to briefly discuss why we do not compare our model's performance against that of other methods, e.g., Casanovo. Our paper’s contribution is that of generalizing to residues not present during training. Thus, we would primarily be interested in comparing these predictions against Casanovo’s performances. However, if we were to construct a fair setting and also remove these residues from Casanovo’s training (and remove the vocabulary), its recall would, by construction, be 0. This is because Casanovo and all other comparable tools cannot predict residues not included in their training sets. Thus, these comparisons do not yield any valuable insight and were purposefully omitted in our paper.
>
> In light of the above general clarification, we respectfully ask the reviewers to revisit their assessment of our paper in the context of zero-shot generalization.
>
> ## Reviewer Revisions
>
> ### Revision 1
>
> Concerning the reviewer’s first revision regarding the comparison between our model and baseline models such as Casanovo, AdaNovo, π-PrimeNovo, ContraNovo, and so on, we hope the above section explains our rationale.
>
> However, similarly to what reviewer 4kEk has rightfully pointed out, we do agree to the fact that we should have included comparisons for the model’s prediction of residues it has seen during training. Unfortunately, the rebuttal time was not enough to generate the values yet. However, we intend to generate these values in teacher forcing mode and add them to the paper once possible.
>
> ### Revision 2
>
> The reviewer’s second point of revision concerning the validation of our model on more PTMs is also valid; however, in practice not easily done. Firstly, we want to reiterate the fact that our paper aims to be a first step towards solving the difficult problem of open DNPS. We want to share a new approach with the DNPS community that we have shown to work in the form of a proof of concept.
>
> In this paper, our initial evaluation on peptides with unseen residues aimed to ensure that the corresponding spectra were comparable to the model’s training data, so as not to confound the evaluation with additional challenges typically associated with modified peptides, such as longer sequences, increased spectral complexity, and altered fragmentation patterns. Therefore, we deliberately decided to use hidden residues from the training data set to provide a proof of concept. Nevertheless, we agree that data containing PTMs (such as phosphorylation) must be considered in future research when practical applicability is aimed for. Further, we chose to only hide 2 compounds, as these were the ones that allowed us to remove all peptides containing them without losing too many PSMs.
>
> ### Revision 3
>
> Concerning the reviewer’s third revision touching on data partitioning. In lines 104f of our paper, we mention that we use the MSV datasplits generated by Casanovo (‘we employ the same train, validation, and test splits used in their study’). Since Casanovo is probably the most established model in the DNPS field, we did not think it would be necessary to reiterate the data partitioning setting used by the original authors. However, we understand that this might be necessary, and we can add a clarifying sentence referring to the original paper for more information.
>
> ### Revision 4
>
> The reviewer’s fourth revision fairly points out that our description of the adversarial learning method could benefit from more details. We will list the reviewer’s specific points here and will add them to the paper once possible.
> (1) We use an alternating update strategy: the discriminator is updated first using frozen encoder outputs, followed by an update of the DNPS model (generator) that includes the adversarial loss. For more details, we point you to our code in the supplement: `gan/model.py`.
>
> (2) The total loss used to train the DNPS model is:
>
> $$
> \mathcal{L}\_{\text{total}} = \lambda\_{\text{reg}} \mathcal{L}\_{\text{reg}} - \lambda\_{\text{adv}} \mathcal{L}\_{\text{D}}
> $$
>
> where $\mathcal{L}\_{\text{reg}}$ is the MSE loss for mass prediction, and $\mathcal{L}\_{\text{D}}$ is the discriminator's binary cross-entropy loss. This corresponds to the composite loss $\mathcal{L}\_{\text{Adv}}$ described in Section 3.3.
>
> (3) The values here were picked by trial and error. This detail should have been included in the paper. The weight $\lambda\_{\text{reg}} = 1$ was statically fixed and we only varied $\lambda\_{\text{adv}}$.
>
> (4) The reviewer is absolutely right to ask for an ablation study, and we agree that this would be a valuable addition to our work. This ablation study would especially strengthen the argument for choosing the weights of the loss terms. Given the short rebuttal period, we have not yet had the opportunity to investigate this.

---

> > ### Comment · Reviewer_sSSm · 2025-08-05
> >
> > Thank you for your rebuttal. Your rebuttal addressed the concerns regarding question 4, but the other content has not been fully resolved.

---

### Official Review · Reviewer_4kEk · 2025-07-01

**Clarity:** 2
**Significance:** 3
**Originality:** 4
**Rating:** 5
**Confidence:** 4

**Summary:**

The authors present a novel method for de novo peptide sequencing which specifically allows for the prediction of arbitrary post translational modifications by predicting the *mass* of the residue instead of the identity of the residue. Test show that with multi task training, the model is able to predict the mass of residues that were unseen during training.

**Questions:**

See the weaknesses and extensions section above.

I am giving this paper an accept - I still believe the weaknesses I discussed should be addressed, but they are fairly minor so I am confident the authors can address them in the rebuttal.

If the authors perform the extension I suggested and it substantially improves results, I would consider raising to a strong accept.

**Ethical Concerns:**

["NO or VERY MINOR ethics concerns only"]

**Limitations:**

yes

**Quality:**

3

**Strengths And Weaknesses:**

## Strengths
This is a really interesting and genuinely novel idea, as far as I know. The choice to predict mass should, in theory, allow for predicting arbitrary modifications and therefore allow interesting generalization. The use of multi-task training with real and experimental data is also interesting and shows good results.

In general, the paper is also fairly clear, and the methodology and results are well explained. There are a few details (enumerated below) which could further improve clarity.

Overall, a good paper.

## Weaknesses
There are a few issues of clarity and missing information that would be good to clear up

### Inline results vs. Table
Results are reported generally inline inside the text. This is common for papers submitting to more bio-focused venues, but I think adopting the table result reporting convention could improve clarity given that it is what readers of NeurIPS papers often expect. Having a single table that reports the various different models trained along with some baselines with a few standard metrics (maybe the average mass error, accuracy of residue calling, and performance on unseen modifications) would help the reader clearly understand what was done and what the effects were.

### Comparison to other models
Results against other models are not reported. This is somewhat ok, given that the authors are testing something new (ability to generalize to unseen PTMs) but there are still comparisons that can be made. The authors even report in the text, "While these recall values were somewhat lower than those reported by classification-based transformer models such as AdaNovo," but do not provide the actual numbers. These should be provided both inline after this statement and in the table suggested above.

### Further discussion of mass prediction vs. residue classification
It is hard for me to believe that mass prediction would perform adequately in the real world due to the similarity in mass between many residues. The impact of this on recall should be stated clearly.

### Missing training and architecture details
The authors should provide (perhaps in the supplement) the architecture and training details for each model. Some of this is provided for the adversarial model, but not the others.

## Extensions
I believe the authors could train a model with multiple objectives for both mass prediction *and* residue classification, with a binary "has PTM" classifier as well. This could allow the best of both worlds, where the model defaults to the residue classifier if there is no PTM (allowing it to distinguish between similar-mass residues) and to mass prediction otherwise.

---

> ### Author Rebuttal · Authors · 2025-07-31
>
> Firstly, we want to thank the reviewers for their constructive comments on our paper. Since we attempted to tackle a difficult problem in DNPS, the community’s feedback is critical to us, and we genuinely appreciate it.
>
> ## Reviewer Concerns
>
> ### Concern 1
>
> The reviewer's first concern regarding inline vs. table results is very valid, and we do recognize that adding a table would add to the clarity of the paper, especially given that NeurIPS readers might expect to have a table. We have added a table containing all model performances below and will add it to the paper once possible.
>
> ### Concern 2
>
> The reviewer's second concern regarding model comparison is also appreciated. As the reviewer mentions, comparing our zero-shot PTM predictions to those of other classification models (such as Casanovo) does not yield valuable insights. However, the reviewer is right, and comparisons to other models in terms of, e.g., AA recall for residues seen during training would add insight, and we should have added those. Because of the short rebuttal, we currently do not have these values in teacher forcing mode for other models, but our own.
>
> ### Concern 3
>
> Concerning the reviewer's third point touching on the model's capability to perform adequately in real-world applications given very similar masses, we do agree and see this as one of the next central problems to address. Since a similar remark was raised by reviewer HY8J, we copy the same answer here:
>
> We have already touched on this in the paper, because we do agree with the reviewer that these metrics are highly important, albeit in a later stage of model development.
>
> Currently, the model would only predict the fewest peptides correctly, giving a very low recall. However, this is to be expected. As we mention in lines 213ff: ‘Predicting scalar masses imposes a stricter requirement for precision: small deviations could lead to mismatches when mapping back to discrete residue tokens. In contrast, models like Casanovo [11] benefited from the flexibility of learned embedding spaces, where similar residues could be placed further apart to ease classification.’ To illustrate this problem, I would like to consider the residues I, N, and D with masses of ~113, ~114, and ~115 Da, respectively. For the model to correctly predict an N, it would have to predict the correct mass within 0.5 Da. Thus, even small mistakes will lead to wrong token predictions and depress the peptide recall and precision.
>
> To address this issue, we also proposed a possible avenue in lines 346ff of the paper: ‘exploring the tradeoff between scalar mass prediction and vector-based encodings.’ For example, one could consider not predicting the scalar mass (thus having the problem of very close AA masses), but rather predicting mass encodings - similar to what is used to encode peaks in the spectrum encoders. These vector encodings could still be mapped to a set of residues, similar to what is done in ContraNovo.
>
> ## Extensions
>
> Firstly, many thanks to the reviewer for taking the time to think about possible improvements. Our paper is specifically intended for people to build upon, and we really hope to inspire future research in this direction. The reviewer’s suggestions seem very much applicable and could indeed work well. Unfortunately, given the short rebuttal time, we have not been able to incorporate these modeling ideas.
>
>
> ## Table
>
> | Real | Sim. Type | GAN | Val MSE | Test MSE Exp. | Test MSE Sim. | Median Abs. Err. Exp. | Median Abs. Err. Sim. | Recall (AA) | Recall M\[Ox] | Recall C\[CAM] |
> | ---- | --------- | --- | ------- | ------------- | ------------- | --------------------- | --------------------- | ----------- | ------------- | -------------- |
> | ✅    |           |     | 96.84   | 160.64        | N/A           | 0.56                  | N/A                   | 62.37%      | 14.18%        | 0.60%          |
> |      | 1         |     | 32.27   | 16265.52      | 10.44         | 126.34                | 2.11                  | 1.53%       | 0.72%         | 0.71%          |
> |      | 2         |     | 40.50   | 3181.56       | 108.96        | 39.42                 | 6.16                  | 7.18%       | 0.44%         | 0.27%          |
> |      | 3         |     | 244.26  | 15648.92      | 263.98        | 121.47                | 4.25                  | 1.46%       | 1.19%         | 0.84%          |
> | ✅    | 3         |     | 174.72  | 239.94        | 175.89        | 3.17                  | 2.95                  | 38.86%      | 14.41%        | 3.18%          |
> | ✅    | 3         | ✅   | 189.86  | 254.82        | 177.30        | 3.63                  | 2.91                  | 37.18%      | 15.11%        | 3.10%          |

---

### Official Review · Reviewer_HY8J · 2025-07-03

**Clarity:** 2
**Significance:** 1
**Originality:** 2
**Rating:** 2
**Confidence:** 4

**Summary:**

This paper presents a novel approach to de novo peptide sequencing by reframing it as a mass regression problem, which enables the model to perform zero-shot identification of PTMs. The transformer-based model is trained using a multi-task learning framework that combines experimental spectra with diverse simulated spectra, and it employs an adversarial scheme to align the representations of these different data types.

**Questions:**

How does the model perform under autoregressive decoding without teacher forcing?

**Ethical Concerns:**

["NO or VERY MINOR ethics concerns only"]

**Final Justification:**

Using teacher forcing during inference is a critical flaw in the methodology, so I keep my scores.

**Limitations:**

yes

**Paper Formatting Concerns:**

No major formatting issues

**Quality:**

1

**Strengths And Weaknesses:**

Strengths
- The author leverages the physical mass of residues as a generalizable feature, which is different from fixed-vocabulary classification and helps in handling novel PTMs.

Weaknesses

Major
- All performance metrics are obtained under teacher forcing. This is a significant limitation that prevents an assessment of the model's real-world de novo sequencing accuracy.
- The end-to-end peptide identification performance (e.g., peptide recall) is not reported.
- The model appears to predict the mass of an unseen methionine oxidation (M+O, ~147.04 Da), but since this is very close to the mass of phenylalanine (F, ~147.07 Da) in the training set, it almost certainly reflects memorization rather than true generalization. This weak claim distracts from the model’s actual capabilities.

Minor
- Figure 2 B: cystein -> cysteine
- Line 108: zero-short -> zero-shot
- Line 157: Mulit -> Multi

---

> ### Author Rebuttal · Authors · 2025-07-31
>
> Firstly, we want to thank the reviewers for their constructive comments on our paper. Since we attempted to tackle a difficult problem in DNPS, the community’s feedback is critical to us, and we genuinely appreciate it.
>
> ## General Remarks
>
> Multiple reviews have commented on two points: the practical applicability of our approach and the missing comparisons to other methods, such as Casanovo.
>
> Regarding the practical utility of our approach, we want to take this opportunity to restate very clearly that we do not claim to have a full model that can be practically applied yet. The problem of open search DNPS is truly hard, as is acknowledged by one reviewer. We believe to have found an approach that could potentially lead the DNPS community (over time) towards true open search models. We appreciate that one reviewer has recognized our novelty. We see our paper as a first step and mentioned this transparently in the paper’s limitations section. As a result, we did not provide peptide-level performance metrics, as our contribution is not the improved prediction of full peptides, but rather to demonstrate a first step towards the generalization to unseen residues. However, we do see that this point can be emphasized more clearly throughout the paper, and we will add further clarification to the camera-ready version once possible.
>
> Secondly, we want to briefly discuss why we do not compare our model's performance against that of other methods, e.g., Casanovo. Our paper’s contribution is that of generalizing to residues not present during training. Thus, we would primarily be interested in comparing these predictions against Casanovo’s performances. However, if we were to construct a fair setting and also remove these residues from Casanovo’s training (and remove the vocabulary), its recall would, by construction, be 0. This is because Casanovo and all other comparable tools cannot predict residues not included in their training sets. Thus, these comparisons do not yield any valuable insight and were purposefully omitted in our paper.
>
> In light of the above general clarification, we respectfully ask the reviewers to revisit their assessment of our paper in the context of zero-shot generalization.
>
> ## Reviewer Concerns
>
> ### Concern 1 & 2
>
> The reviewer's first two concerns touch on the use of teacher forcing and peptide-level metrics that we do not report. Our choice to use teacher forcing for evaluation statistics is hopefully explained above, as well as why we do not report peptide-level performances.
>
> ### Concern 3
>
> The reviewer's last point concerns the mass similarity of M[Ox] and F, which only differ in mass by about 0.03 Da. This mass similarity might cause memorization, instead of generalization. And we do mention this transparently in lines 227ff of our paper: ‘However, this perhaps reflected a memorization artifact: the mass of methionine oxidation (∼147.04 Da) was only ∼0.03 Da less than phenylalanine (F), a residue included in training, indicating the model might simply be reproducing familiar masses. The observed bias toward masses seen during training was consistent with trends in generalized zero-shot learning.’ Nonetheless, this is indeed a valid concern that we have to think about, and the reviewer is right in pointing it out. As mentioned in the paper, a bias towards masses present during training is expected, which is why we purposefully evaluated the model’s performance on C[Carbamidomethyl] too, which only has the next closest mass of about 3 Da (Y).
>
> ### Others
>
> Additionally, we want to thank the reviewer for pointing out the spelling mistakes. These will, of course, be fixed before publication.
>
> ## Reviewer Questions
>
> Lastly, regarding the reviewer's question on model performance when running in autoregressive mode. As mentioned before, we do not think this metric is meaningful for the current state of the model. However, we want to outline the issues of the current design and what we believe a future point of improvement could be. We have already touched on this in the paper, because we do agree with the reviewer that these metrics are highly important, albeit in a later stage of model development.
>
> Currently, the model would only predict the fewest peptides correctly, resulting in very low recall. However, this is to be expected. As we mention in lines 213ff: ‘Predicting scalar masses imposes a stricter requirement for precision: small deviations could lead to mismatches when mapping back to discrete residue tokens. In contrast, models like Casanovo [11] benefited from the flexibility of learned embedding spaces, where similar residues could be placed further apart to ease classification.’ To illustrate this problem, we would like to consider the residues I, N, and D with masses of ~113, ~114, and ~115 Da, respectively. For the model to correctly predict an N, it would have to predict the correct mass within 0.5 Da. Thus, even small mistakes will lead to wrong token predictions and depress the peptide recall and precision.
>
> To address this issue, we also proposed a possible avenue in lines 346ff of the paper: ‘exploring the tradeoff between scalar mass prediction and vector-based encodings.’ For example, one could consider not predicting the scalar mass (thus having the problem of very close AA masses), but rather predicting mass encodings - similar to what is used to encode peaks in the spectrum encoders. These vector encodings could still be mapped to a set of residues, similar to what is done in ContraNovo.

---

> ### Comment · Reviewer_HY8J · 2025-08-05
>
> I thank the authors for addressing my concerns; however, I remain convinced that teacher forcing fails to reflect real-world de novo sequencing, rigorous evaluation under fully autoregressive decoding is essential to validate any claims of generalization.

---

### Decision · Program_Chairs · 2025-09-17

**Decision:**

Reject

**Comment:**

This paper proposes a novel reformulation of de novo peptide sequencing as a mass regression task, enabling zero-shot prediction of unseen PTMs through adversarial multi-task learning on experimental and simulated spectra. The idea is original and addresses an important challenge in proteomics, with potential to move beyond fixed-vocabulary DNPS models. The authors are transparent about the work being an early proof of concept.

However, reviewers raised consistent concerns about methodology and evaluation. In particular, reliance on teacher forcing during testing undermines claims of generalization, no systematic comparisons to baseline models are provided, and validation is restricted to two common PTMs with very low recall. Missing details on adversarial training and the absence of peptide-level metrics further weaken the contribution. Overall, while the direction is promising, the submission is too preliminary for NeurIPS.